# Diseases from the Spectrum of Dermatitis and Eczema: Can “Omics” Sciences Help with Better Systematics and More Accurate Differential Diagnosis?

**DOI:** 10.3390/ijms241310468

**Published:** 2023-06-21

**Authors:** Radoslaw Spiewak

**Affiliations:** Department of Experimental Dermatology and Cosmetology, Faculty of Pharmacy, Jagiellonian University Medical College, ul. Medyczna 9, 30-688 Krakow, Poland; radoslaw.spiewak@uj.edu.pl

**Keywords:** dermatitis, eczema, systematics, clinical criteria, molecular diagnostics, genomics, transcriptomics, proteomics, lipidomics, metabolomics

## Abstract

Researchers active in the field of inflammatory skin diseases from the spectrum of dermatitis and eczema are well aware of a considerable overlap in the clinical pictures and proposed sets of diagnostic criteria for these diseases, which can hardly be overcome through the clinical or epidemiological research. In effect, patients are included in studies based on vague and overlapping criteria, while heterogeneous study populations may, in turn, lead to non-representative outcomes and continued confusion. In this narrative review, a systematics of diseases from the spectrum of dermatitis and eczema is proposed based on the origins of causative factors and the pathomechanisms involved. Difficulties in differentiating between these diseases are discussed, and the extent to which advances in the “omics” sciences might help to overcome them is considered. Of all the “omics” research in this field, more than 90% of the published papers were devoted to atopic dermatitis, with a striking underrepresentation of other diseases from the spectrum of dermatitis and eczema, conditions which collectively exceed the rates of atopic dermatitis by far. A greater “omics” research effort is urgently needed to tackle other dermatitides, like allergic, irritant and protein contact dermatitis, as well as radiation, seborrheic, stasis or autoimmune dermatitis. Atopic dermatitis findings should be validated not only against healthy donors but also other dermatitides. A clinic-oriented approach is proposed for future “omics” studies in the field of dermatitis and eczema.

## 1. Introduction

Dermatitis (eczema) is a noninfectious inflammation of the epidermis and dermis that manifests itself through an array of efflorescences, including erythema, edema, inflammatory infiltrate, papules, vesicles, scales, serous crusts and lichenification. Depending on the stage and intensity of the disease, these skin changes may emerge in various constellations simultaneously or may evolve from one another. They are typically accompanied by the subjective sensation of itch (pruritus), pain or stinging or burning sensations of various intensities. The histological picture of dermatitis/eczema includes spongiosis, acanthosis, parakeratosis or hyperkeratosis in the epidermis, in addition to lymphocytic and granulocytic infiltration of the upper dermis and epidermis. Spongiosis is present in all diseases manifesting clinically as eczema; however, there are diseases with spongiosis that do not belong to the eczema spectrum, e.g., pityriasis rosea, erysipelas or tinea [1]. The term “dermatitis” means literally “inflammation of the skin”; however, its use in dermatology is restricted to a subgroup of non-infectious inflammatory skin diseases with similar clinical appearances. Therefore, tinea (a fungal infection of the skin), psoriasis or inflammatory acne will not be referred to as “dermatitis” even though inflammation of the skin is their inherent feature. Depending on the country and dermatology school, the term “eczema” could refer to acute dermatitis (e.g., in the USA) or chronic dermatitis (e.g., in Germany). Moreover, some authors maintain that “eczema” means dermatitis with a known cause, while “dermatitis” would suggest that there is no definite diagnosis yet. Finally, the term “eczema” is used by some authors as a synonym of “atopic dermatitis” (AD), which should be strongly discouraged as misleading and confusing. AD is just one of many diseases on the spectrum of dermatitis and eczema (SoDE). In light of these contradictions, it seems most reasonable to consider the terms “dermatitis” and “eczema” synonyms. In the present article, both terms will be used interchangeably.

## 2. Present Diagnostic Difficulties

The terms “dermatitis” and “eczema” do not refer to any specific disease but a constellation of clinical features common to a heterogeneous group of diseases. Diseases from the SoDE share very similar appearances—at some stage, they all manifest erythema, edema, infiltration and scaling of the skin. Edema, inflammatory papules, vesicles or exudate (oozing) are typically seen in the acute phase, while skin dryness, cellular infiltrate, scaling, hyperkeratosis or fissuring are predominant in the chronic phase. Eczemas are accompanied by pruritus or burning or stinging sensations of various intensities [2,3]. Despite apparent clinical similarities, diseases from the SoDE may have different etiologies, courses and prognoses [4,5]. Accordingly, they may require different therapeutic strategies and patient management [6,7,8,9,10,11,12]. The similarity in clinical pictures poses a considerable diagnostic challenge for the clinician; moreover, there is a considerable risk of developing a secondary dermatitis of a different etiology in the course of the primary dermatitis [13,14]. For example, a patient may develop a contact allergy (synonym: type IV allergy) to skincare products or topical drugs used in the treatment of AD, resulting in a secondary allergic contact dermatitis (ACD). A sequence of various dermatitides that appear almost identical is also possible, e.g., in growing child, AD may first be complicated and later replaced by ACD without the patient or doctor noticing the change. The similarity of symptoms and the causal heterogeneity of diseases from the SoDE makes it difficult to create a systematics with unequivocal diagnostic criteria. As a result, more than 60 variants of eczema are mentioned in dermatology textbooks, some of which may sound exotic, e.g., “eczema autotoxicum”, while others are quite popular, although misleading, e.g., “neurodermatitis”, “milk crust” or “eczema infantum”.

An accurate diagnosis is a prerequisite for optimal patient management. Until now, the diagnosis of diseases from the SoDE was founded on the patient’s medical history and clinical picture. Unfortunately, the diagnostic criteria used in this process were rather vague and arbitrary, and there were also considerable overlaps between the criteria for different diseases. For example, eczema located in flexural areas is popularly considered a hallmark of AD; nevertheless, flexural predominance is observed in many skin diseases [15,16,17]. Structural examinations, including histology and the immunophenotyping of skin specimens, also do not guarantee an ultimate diagnosis [18]. Under such circumstances, diagnoses may be biased by the experiences and individual beliefs of the diagnosing doctors. With clinical features insufficient for establishing an accurate diagnosis, a search for molecular markers that would effectively distinguish between various entities is warranted. Unfortunately, the above-mentioned difficulties in clinical diagnosis may also hamper progress in molecular research: patients recruited into studies based on vague and overlapping clinical criteria would arguably form heterogeneous populations with research outcomes that are not truly representative of any specific disease from the spectrum, an issue which is discussed below using the example of AD. Typically, clinicians recruit patients for the studies, and molecular scientists have no choice but rely on their judgement. Fortunately, these limitations may be overcome by clustering patients with similar molecular profiles and investigating distinctive clinical features they might share. This kind of “reverse engineering” would probably redraw borders between clinical entities within the SoDE and redefine diagnostic criteria. In the search for reliable markers, the “omics” approach seems favorable because large arrays of candidate markers must be effectively tested in all diseases from the SoDE via high-throughput methods. Unfortunately, of all the “omics” research published to date in the field of dermatitis and eczema, 90.4% of papers in general and 97.6% of papers in the last 5 years were devoted to AD only (Appendix A). A striking underrepresentation of other diseases from the SoDE becomes apparent when comparing the above statistics with the prevalence data: in a study carried out in five European countries, AD was reported in 7.8% of the general adult population, contact dermatitis (irritant or allergic) in 15.5% and other eczemas in 14.0% [19]. A wisely planned and concerted action would be necessary in order to fill in the gaps and cover the entire SoDE.

## 3. How to Classify Diseases from the Spectrum of Dermatitis and Eczema (SoDE)

Table 1 presents the author’s attempt at ordering dermatitides according to their probable etiology and pathomechanisms—the features that determine the clinical course, management and prognosis of a disease and translate into the patient’s quality of life. The author has used this systematics for more than 15 years in both teaching and routine clinical practice while assigning diagnostic and therapeutic modalities in individual patients. The founding idea behind it was to encourage clinicians to think about the possible causes and mechanisms involved in each individual case rather than applying a pattern of thinking along the deceptive line of “it looks like X, thus it is X”. Avoiding this type of shortcut also seems essential when planning the future of “omics” research on diseases from the SoDE.

From Table 1, one might notice that some entities, though seemingly closely related, actually fall into different classes of etiopathology. For example, the term “contact dermatitis” is actually an umbrella term covering different diseases, including irritant contact dermatitis (ICD), its variant phototoxic dermatitis, ACD, its variants photoallergic contact dermatitis and systemic reactivation of ACD (SRACD), and finally, protein contact dermatitis (PCD) [22]. The common feature of contact dermatitides is the means of exposure to provoking agents—in all contact dermatitides, the triggering factors enter the skin from the outside after direct contact (the exceptions are SRACD and systemic photoallergic dermatitis, see below). This shared exposure route is reflected in the similarities of their clinical pictures; however, the physicochemical natures of the causative factors and the underlying pathomechanisms are different in each entity. The cases of intrinsic and extrinsic AD illustrate the heterogeneity of AD, which will be discussed in the section devoted to this disease.

In a clinical routine, it may prove practical to use collective terms that combine diseases that differ in their etiopathologies yet share relevant features, like clinical appearance, route of exposure or legal status. Examples of such collective terms are summarized in Table 2. These auxiliary diagnoses will not be further discussed because each individual case can be assigned to a class in Table 1, e.g., airborne dermatitis may be either ACD, ICD, PCD or extrinsic AD.

## 4. An Overview of Diseases from the Spectrum

The following overview of the diseases from the SoDE is focused on clinical and diagnostic features relevant to present and future “omics” research. In order to ascertain their compatibility with current terminology, they will be organized along diagnoses currently used in clinical routine and molecular research, with a discussion of possible limitations resulting from such an approach.

### 4.1. Atopic Dermatitis (AD)

With more than five thousand dedicated articles indexed in PubMed in the last 5 years, AD (synonym: atopic eczema) is certainly one of most studied skin diseases. The accumulated knowledge on AD exceeds the scope of the present article by far; therefore, the reader is referred to state-of-the art reviews covering general aspects [23,24], the newest treatment options [25,26,27], comorbidities [28,29], the role of skin microbiota [30,31] and a systematic review of expert guidelines on the diagnosis and management of AD [32] to name just a few recent examples. This section will focus only on the possible application of “omics” sciences in the improvement of the diagnostic criteria and differential diagnosis of AD versus other diseases from the SoDE.

AD seems to be a most popular diagnosis within the SoDE. Owing to its strong presence in social consciousness and on social media, the frequent use of the term in cosmetic marketing and its popularity with both patients and doctors, AD is arguably also one of the skin diseases most prone to overdiagnosis nowadays. The classical diagnostic criteria of AD proposed by Hanifin and Rajka [33], which are still the most frequently cited and used to date, are a collation of clinical features that can each be found in other diseases and when looking at constellations of symptoms. This leaves a considerable space for a clinician’s subjective judgement and may partly explain the striking differences in the reported prevalence rates of AD in various countries, which range from 0.18 to 38.33% (median 4.91%) [2]. The creators of the above-mentioned criteria seem to have been aware of these limitations. As Jon Hanifin put it, “until there is a precise laboratory marker, we are dependent on obviously imprecise clinical criteria. We must constantly remember that these criteria are imprecise; no amount of mathematical, statistical manipulation or validation will make them precise” [34]. The present name of the disease, “atopic dermatitis”, implies a causative role of atopy in the disease, which has long been disputed. Georg Rajka rated the name “atopic dermatitis” as “an unfortunate choice of term”, further explaining that “the flaw lies in the conclusion from recent experience that the disease can no longer be considered a typical atopic disease” [35]. In an attempt at reconciling fire with water, the European Academy of Allergy and Clinical Immunology (EAACI) proposed the term “Atopic Eczema/Dermatitis Syndrome (AEDS)” as a replacement for the term “atopic dermatitis”. According to the proposal, AEDS was divided into “nonallergic AEDS” and “allergic AEDS”, with the latter further subdivided into “IgE-associated allergic AEDS” and “non-IgE-associated allergic AEDS”. In the same paper, atopy was defined as a “personal or familial tendency to produce IgE” [36]; thus, the term proposed in the classification, “non-IgE-associated allergic AEDS”, would literally expand to “non-IgE-associated, i.e., non-atopic, allergic atopic, i.e., characterized by tendency to produce IgE, dermatitis/eczema syndrome”, a term implicating the absence and at the same time the presence of an association with IgE. This somewhat bizarre story illustrates the deep confusion around the disease currently known as “atopic dermatitis”, a confusion that may undermine progress in both clinical and molecular research. In an attempt at solving this conundrum, Bos et al. have proposed that only AD cases with allergen-specific IgE as a hallmark of atopy (corresponding with extrinsic AD, see below) should be referred to as “atopic dermatitis”, while the variant without atopy (intrinsic AD) should be called “atopiform dermatitis”, hinting at the possibility that these entities may in fact constitute two separate diseases [37]. This pertinent proposal did not gain any wider recognition. Perhaps substituting the name “Hanifin Rajka Syndrome” for AD could offer a less confusing and more acceptable interim solution until we know better. The proposed term builds on the widely recognized Hanifin and Rajka criteria and comprises all cases fulfilling these criteria without speculating on the underlying causes and mechanisms. The term “syndrome” also implies that one deals with a group of symptoms rather than a disease with defined causes and a pathophysiology [38].

Studies on AD demonstrate a considerable variability in genetics, epidemiology, clinical phenotypes and treatment responses [39]. Racial differences in the mechanisms involved have led to calls for dividing AD into Asian AD, African AD and European AD [40,41]. A widely accepted division of AD is its division into extrinsic AD and intrinsic AD. Extrinsic AD (exogenous eczema) is characterized by skin barrier impairment, mainly due to filaggrin mutations, and the development of specific IgE to common environmental allergens, e.g., house dust mites, fungal spores, airborne pollen or foods [41]. There is growing evidence that the production of specific IgE and the development of a type I allergy toward environmental allergens is a phenomenon secondary to skin barrier impairment and inflammation [42]. The intrinsic subtype of AD (endogenous eczema) is characterized by a predominance of the female gender, normal skin barrier function and no excess of type I allergies. Instead, an increased prevalence of metal allergies is observed in this group, with suprabasin deficiency as a possible predisposing factor [41]. It is not clear, however, whether the reported excess in metal allergies is a direct consequence of suprabasin deficiency, an event secondary to the skin inflammation (danger signals facilitating the induction of an contact allergy), a confounder due to the higher prevalence of nickel allergies among women or perhaps SRACD to nickel mimicking AD. With the name implying a causal role of atopy (an IgE-mediated disorder), available data on the therapeutic effectiveness of anti-IgE treatment in AD are surprisingly scarce and contradictory. Overall, they suggest that AD patients would benefit more from other treatment modalities; however, a fraction of the patients seems to benefit from anti-IgE treatment [43,44,45,46]. As anti-IgE treatment targets a well-defined link in the chain of atopic (type I) allergic reactions, the mixed response once again evokes the question whether these are variants of one disease or rather different diseases with separate underlying mechanisms.

Recent molecular research demonstrated different mechanisms involved in individual patients bearing a diagnosis of AD. Based on a serum biomarker analysis, Thijs et al. divided adult AD patients into four distinct clusters: the patients in cluster 1 were characterized by symptoms with high levels of severity, the involvement of large body surfaces and the highest levels of the pulmonary and activation-regulated chemokine (PARC/CCL18), tissue MP1 inhibitor and soluble CD14 antigen. Cluster 2 was characterized by a low level of clinical severity and the lowest levels of IFN-alpha, tissue MP1 inhibitor and vascular endothelial growth factor (VEGF). The patients grouped into cluster 3 demonstrated high levels of severity and the lowest levels of IFN-beta, IL-1, epithelial cytokines IL-25 and IL-33 and thymic stromal lymphopoietin (TSLP). Patients in cluster 4 shared low levels of disease severity and the highest levels of IL-1, IL-4, IL-13 and TSLP [47]. In a more recent study of another population with AD, the same research group developed a new division that only partly overlapped with the previous results: in the newer division, cluster A comprised patients with higher levels of skin-homing C-C chemokines (CTACK/CCL27, TARC/CCL17, MDC/CCL22 and RANTES/CCL5) and IL-1R1; this “skin-homing chemokines/IL-1R1–dominant” cluster did not correspond to any cluster from the earlier division. Cluster B represented patients with the highest levels of TH2-related (IL-4, IL-5 and IL-13), TH1-related (IFN-gamma, TNF-alpha and TNF-beta), TH17-related (IL-17 and IL-21) and epithelial cytokines (IL-25, IL-33 and TSLP). This “TH1/TH2/TH17-dominant” cluster B was comparable to cluster 4 in the earlier division. Cluster C comprised patients with high levels of TH2-related cytokines (PARC, IL-13, IL-5, eotaxin and eotaxin-3), IL-22 and IL-33. The constellation of molecular markers in the “TH2/TH22/PARC-dominant” cluster C corresponded with cluster 1 from the previous work. Cluster D represented patients with AD characterized by a relatively low inflammatory state which made them distinctive from other clusters due to low serum levels of TH2/severity-related (MDC, PARC and TARC) and eosinophil-related markers (RANTES, eotaxin and eotaxin-3). The secretory profile of this “TH2/eosinophil-inferior” cluster D resembled the previously identified cluster 2 [48]. A similar approach applied by the same group in a pediatric AD population revealed four pediatric clusters: children stratified in cluster 1 (“TH2 cell/retinol–dominant”) were characterized by the highest levels of retinol-binding protein 4 (RBP4) and with elevated levels of IL-4, IL-5, IL-13 and TSLP. Cluster 2 (“skin-homing–dominant cluster”) consisted of children with the highest levels of apelin and markers related to skin homing (PARC/CCL18, TARC/CCL17 and CTACK/CCL27) and the lowest levels of markers related to tissue remodeling and angiogenesis (adiponectin, MMP-8 and TIMP1). This cluster also had the highest incidences of food allergies. Cluster 3 (“TH1 cell/TH2 cell/TH17 cell/IL-1–dominant”) was defined by the highest levels of biomarkers related to the TH1 cell pathway (IL-2, IL-12, IFN-alpha, IFN-gamma, TNF-alpha, TNF-beta, MIG/CXCL9 and ITAC/CXCL11), the TH2 cell pathway (IL-4, IL-5, IL-13, eotaxin-3/CCL26, TSLP and MCP-4/CCL13), the TH17 cell pathway (IL-23, IL-26, MIP3a/CCL20 and GM-CSF), the IL-1 family pathway (IL-1a, IL-1Ra, IL-1R1, IL-18BPa and IL-37), the TNF superfamily pathway (TNFR1, TNFR2, TWEAK/TNFSF12 and LIGHT/TNFSF14) and T-cell activation (sIL2Ra). Cluster 4 (“TH1 cell/IL-1/eosinophil–inferior cluster”) comprised children with the highest levels of the chemokines RANTES/CCL5 and PF4/CXCL4 and the monocyte activation marker soluble CD14, as well as the lowest levels of biomarkers related to the TH1 cell pathway (MIG/CXCL9, ITAC/CXCL11 and MIP1b/CCL2), eosinophil trafficking (eotaxin-1/CCL11 and eotaxin-3/CCL26), the IL-1 family pathway (IL1R1 and IL-18BPa), the TNF superfamily pathway (TNFR1, TNFR2 and TWEAK/TNFSF12), neutrophil activation and trafficking (elastase and GCP2) and T-cell activation and skin homing (sIL2Ra and CTACK). The incidences of food allergies were lowest in this cluster. Of all the pediatric clusters, only cluster 3 (TH1 cell/TH2 cell/TH17 cell/IL-1–dominant) corresponded with a cluster previously defined in adults, i.e., cluster B (cluster 4 in the study of 2017) [49]. The above-mentioned studies are exemplary with respect to application of an “omics” methodology in future research in this field. At the same time, they raise a question as to what extent the heterogeneity of the results was due to imprecise clinical criteria and the resultant heterogeneity of the populations studied. Nevertheless, these studies illustrate the change in AD research thanks to “omics” studies, though the real meaning of these discoveries can be only assessed in the broader context of other diseases from the SoDE and preferably other skin diseases. A recent review of the perspectives on improving the diagnosis of AD due to advances in “omics” research offers a detailed overview of studies published prior to its completion [50]. Table 3 collates further “omics” studies that were not covered by the previous review with the possible implications of their results on the diagnosis of AD.

A main conclusion from the “omics” studies published to date is that molecular methods have been applied mainly to understand the pathomechanisms of AD, while differences between AD and other skin diseases have been rarely studied, and no study published to date compares AD with other diseases from the SoDE. Therefore, it is not clear which of the markers indicated in the past research are typical of a specific diagnosis and which are expressed in various diseases from the SoDE or in other inflammatory skin diseases.

### 4.2. Irritant Contact Dermatitis (ICD)

ICD is an acquired inflammatory disease of the skin provoked by irritants, i.e., exogenous physical or chemical agents causing damage to keratinocytes. Examples of irritants are detergents, solvents, degreasers, dry air, low temperature, acids and alkalis, repetitive microtrauma, pressure and friction [66]. Even factors essential to life like water or foods may cause ICD under prolonged exposure. The potency of a chemical irritant and its ability to penetrate the skin are determined by its properties, including its molecular weight, ionization state and fat solubility. Different irritants target different structures in the epidermis depending on their physicochemical properties [67]. In acute ICD, strong irritants may cause visible skin symptoms within minutes or hours. There is a quantitative rather than qualitative difference between acute ICD and a chemical burn. In a chemical burn, the destruction of the epidermis and dermis with erosions, ulceration and necrosis may develop in areas of maximal exposure to the irritant, while the surrounding skin may manifest signs of acute ICD, including edema, dark-red or livid-red erythema, a glazed, parched or scalded appearance and serous exudate in case of a deeper-reaching destruction of epidermis. These symptoms are accompanied by pain or a burning or stinging sensation (typically, there is no urge to scratch) [3]. A typical clinical picture of chronic ICD consists of hyperkeratosis with large, thick scales and a tendency toward the fissuring (breaking) of the thickened skin because of its dryness and stiffness. ICD does not involve specific hypersensitivity reactions, meaning that all people exposed to the irritants will develop similar skin reactions, though some people will succumb to a lower dose or intensity, while others (“thick skinned” individuals) will react to a higher intensity of the damaging factor. Chronic ICD may result from repeated subthreshold damages, i.e., insults by various irritants that individually go unnoticed, but when the interval between the insults is too short to allow for complete recovery, these combined effects may lead to noticeable skin disease [67]. This cumulative effect may be due to various irritants damaging the skin sequentially or in parallel. Therefore, unlike ACD, clinical improvement in ICD may be expected only if all irritants in the patient’s surroundings (work, home, hobby, climate, etc.) are reduced.

To date, only one “omics” study has offered some insight into the mechanisms of ICD. A lipidomics analysis of skin samples from SLS-induced experimental ICD showed a significant increase in N-acyl ethanolamides (NAE), palmitoyl and stearoyl ethanolamides and 12-hydroxyeicosatetraenoic acid levels when compared with UV-induced erythema in the same volunteers [68]. However, it is not clear whether the observed differences were characteristic of ICD or perhaps specific to the mode of action of the SLS (sodium lauryl sulfate), which is known to interfere with the production and processing of epidermal lipids [69].

### 4.3. Phototoxic Dermatitis

Similar to ICD, the damage to the epidermis in phototoxic dermatitis depends on the physicochemical properties of the phototoxic agents, bearing similarities with ICD. This damage is caused by radicals generated during photochemical reactions, meaning that the development of phototoxic dermatitis is always preceded by exposure to light (typically UVA) in the presence of a photosensitizer, i.e., a chemical that absorbs the energy of light. Photons caught by a photosensitizer’s molecule increase its internal energy by pushing its electrons into higher orbitals. This excess chemical energy may facilitate the creation of free radicals (type 1 phototoxic reactions) or oxygen radicals (type 2) that damage cell structures, e.g., lipid membranes or proteins. In an excited state, some photosensitizers (e.g., psoralens) are capable of creating stable covalent bonds with DNA strains [70]. These structural damages lead to a release of inflammatory mediators, complement activation, granulocyte migration and keratinocyte apoptosis, which manifest clinically as dermatitis. No specific (adaptive) immune response is involved in phototoxicity, differentiating it from photoallergic reactions, though there are chemicals that may act as both phototoxic and photoallergic agents [71]. Phototoxic agents may enter the skin from outside, e.g., ingredients in cosmetics or topical drugs or from plants (phototoxic contact dermatitis) or from inside, e.g., drugs or dietary supplements (systemic phototoxic dermatitis) [72].

### 4.4. Radiodermatitis (RD)

RD (synonym: radiation dermatitis) is another form of inflammatory skin disease caused by an external factor: ionizing radiation. Individual susceptibility seems to play no role or a limited role because all exposed people will develop RD depending on the dose of energy absorbed. Acute RD develops within 90 days of irradiation and initially presents as primary transient erythema, followed by generalized erythema, pruritus, xerosis, hyperpigmentation, dry scaling and peeling of the skin. Moist scaling hints at deeper damage to the skin with the exudation of tissue fluids. These changes may be accompanied by hair loss (anagen effluvium) in the irradiated area. Dermatitis that emerges (or persists) more than 90 days since the last bout of irradiation, which is referred to as chronic RD, may manifest as skin thinning due to the atrophy of the dermis and epidermis or skin thickening of the dermis due to fibrosis and may be accompanied by edema, dyspigmentations, telangiectasias or dermal necrosis [73]. Ionizing radiation causes extensive and irreversible genetic damage to nuclear and mitochondrial DNA that inhibits cells’ ability to replicate. It is especially detrimental to keratinocytes in the Malpighian layer where all the mitotic activity of the epidermis takes place. This DNA damage, combined with the generation of reactive oxygen species (ROS) that damage structural proteins, enzymes and lipid membranes, initiates epidermal and dermal inflammatory responses and skin cell necrosis, which collectively manifest as RD. ROS damage β-catenin and E-cadherin, which are pivotal proteins in adherens junctions—the cell-to-cell connections that ensure the integrity of the epidermis. Damage to adherens junctions leads to a loss of contact between keratinocytes and spongiosis, which correlates with the severity of the disease. In humans, spongiosis was observed after a 4-week course of radiotherapy with a cumulative dose of 46 Gy, in addition to the upregulation of the Hippo signaling pathway, which seems to be involved in cell proliferation and repair [74]. The role of Hippo pathway activation in dermatitides seems to have not been studied much thus far. Damage to β-catenin and E-cadherin is probably not restricted to RD but a more common occurrence among diseases from the SoDE, all of which include spongiosis [75].

### 4.5. Seborrheic Dermatitis (SD)

SD (synonym: seborrheic eczema) is an inflammatory skin disease confined to regions with high densities of sebaceous glands: the scalp, face, central upper back and sternum [76]. The maturation of the sebaceous glands seems to be a prerequisite for the development of SD; therefore, this disease typically develops after puberty except for infantile SD, when the glands are upregulated by maternal sex hormones. Although the disease’s name suggests an association with the overproduction of sebum (seborrhea), SD may also develop in people with normal sebum outputs [77]. It seems that changes in the sebum composition may also be a contributing factor [78]. Changes in the amount or composition of sebum, in addition to a defective epidermal barrier, amount to the primary events in the pathology of SD that provide favorable conditions for a secondary overgrowth of lipophilic yeasts from the genus Malassezia, which provokes an inflammatory response in the skin [79]. Malassezia spp. are commensal lipophilic yeasts belonging to normal skin microbiota which may turn into opportunistic pathogens under favorable conditions. The hydrolysis of sebum by Malassezia yeasts liberates oleic acid, which possesses irritant properties and activates the innate immune system via pattern recognition receptors, inflammasome, IL-1β and NF-κB, resulting in the secretion of proinflammatory cytokines IL-8, IL-17 and IL-4 [79]. In this way, SD fits into the spectrum of ICD rather than fungal infections, regardless the role of live fungus in its pathology. Malassezia in SD is a source of irritants rather than an invader. Nevertheless, antifungal treatment is a relevant therapeutic option as Malassezia loads correlate with the severity of SD, and a reduction in the Malassezia load leads to a remission of the disease [80]. Ketoconazole—one of the preferred antifungals used in the therapy of SD—also modifies the lipid profile of sebum and suppresses inflammation; therefore, the therapeutic effect of the drug may be not entirely due to its antifungal activity [81].

Infantile SD is a common, self-limiting and benign condition. Its prevalence is the highest in the first 3 months of life and rapidly decreases thereafter [82]. Meanwhile, infantile SD can easily be misdiagnosed as AD. The presence of pruritus, a positive family history of atopy and the age of onset are altogether of limited value in the differential diagnosis between AD and infantile SD. The most distinctive features are the presence of lesions on the forearms and shins in AD, while the localization of skin lesions solely to the napkin area or the axillae favors a diagnosis of SD; however, the significance of these features decreases with the spread of lesions to multiple body sites [83]. Once again, this illustrates the need for reliable diagnostic criteria beyond clinical features. A transcriptomic analysis of scales from lesional skin (dandruff) in SD revealed a strong upregulation of the genes coding interleukin-1 receptor antagonist gene (IL-1Ra) and IL-8, as well as the genes S100A8, S100A9 and S100A11 [84]. This study singled out genes whose expression clearly differentiated between involved and uninvolved skin, as well as between SD patients and healthy controls; however, it is not known which of those would differentiate SD from other diseases in the SoDE. The proinflammatory cytokine IL-8 is probably upregulated in every dermatitis, limiting its value in the differential diagnosis. On the other hand, the suppression of genes related to lipid metabolism observed in the aforementioned study may correspond with the assumption that changes in the lipid composition of sebum play a role in the development of SD, making these genes an interesting target for further research. An increased risk of developing SD may also be connected with the human leucocyte antigen (HLA) alleles A*32, DQB1*05 and DRB1*01, mutations in the LCE3 gene cluster and mutations impairing the ability of the immune system to restrict Malassezia growth [79].

### 4.6. Allergic Contact Dermatitis (ACD)

ACD (synonym: allergic contact eczema) is an inflammatory skin disease initiated by specific immune reaction to a hapten. Haptens are low-molecular-weight chemicals that are not immunogenic per se. Instead, these reactive chemicals can bind to the body’s own proteins in a process referred to as haptenization. As a result of haptenization, the spatial conformation of endogenous proteins becomes distorted by the covalent bonds with haptens to such extent that they no longer are recognized as self-antigens and can provoke an immune response like any non-self-antigen [85]. The sensitizing potential of a hapten is determined by its physicochemical properties [86,87]. The most frequent causes of ACD are metals and cosmetic ingredients, mainly fragrances and preservatives. Not all environmental substances responsible for ACD are actual haptens—some of them are hapten precursors that convert into haptens in one of two possible ways: prehaptens turn into reactive haptens due to spontaneous oxidation via contact with air, while prohaptens undergo enzymatic activation in the host’s organism [88]. For example, the fragrance linalool is a prehapten that spontaneously degrades into hydroperoxides, which are the actual sensitizing haptens [89]. Another fragrance, cinnamic alcohol, is a prohapten that is transformed into the actual sensitizer cinnamic aldehyde in an enzymatic process involving alcohol dehydrogenase (ADH) and cytochrome P2E1 (CYP2E1) with NADP+ as cofactor [90].

Only a minority of people exposed to a particular hapten will develop ACD, meaning that individual predisposition is prerequisite for developing a specific immune hypersensitivity: a contact allergy (a delayed-type reaction). The term “contact allergy” (CA) is not synonymous with ACD. Confusing CA with ACD appears to be a quite frequent mistake in clinical practice that sometimes also contaminates published research and may interfere with scientific progress in the field. The term “contact allergy” refers to a state of altered immune response to a specific hapten, which is not synonymous with the disease. Some people with confirmed CA may avoid the development of ACD by keeping away from the causative hapten. Among people who become symptomatic, a majority will develop ACD, while other diseases resulting from CA include allergic contact stomatitis, conjunctivitis, vaginitis, rhinitis and asthma, as well as intolerance reactions to orthopedic prostheses, dental implants and other medical devices.

The natural course of a CA and ACD is divided into an initial sensitization phase and a subsequent elicitation phase [91]. The sensitization phase (synonyms: afferent or induction phase) is facilitated by cells of the innate immune system, beginning with the detection of haptenated proteins by Langerhans cells (LC). Depending on the nature of the hapten, e.g., its irritant potential and the presence of cofactors, e.g., inflammatory cytokines or reactive oxygen species (ROS) at the site of encounter, LCs may ignore the hapten or become activated and carry epitopes (antigenic fragments) of the haptenated proteins to local lymph nodes. Within the lymph node, LCs present the epitopes in the context of major histocompatibility complexes to scores of naïve CD4+ and CD8+ cells, including T helper (Th) type 1 cells. When T cells with TCR receptors fitting to the LC complex are found, they undergo clonal expansion into a population of effector T cells that migrate back to the entry point of the hapten, where they orchestrate an inflammatory reaction, engaging eosinophils, neutrophils, macrophages or cytotoxic lymphocytes. The cell composition of the inflammatory infiltrate determines the clinical appearance of skin lesions. The inflammatory activities of these effector cells results in the death of keratinocytes, which manifests as spongiosis. The types of spongiosis observed in ACD include eosinophilic spongiosis (eosinophils within the foci of spongiosis), spongiosis with subepidermal edema (dermal type ACD), pityriasiform spongiosis (small vesicles with lymphocytes, histiocytes and LC) or haphazard spongiosis (no particular pattern) [92]. When the inflammatory reaction subsides, a fraction of effector cells will turn into resident epidermal or circulating effector memory T cells on standby for another encounter with the hapten, while redundant cells will commit apoptosis. Subsequent exposures to the hapten (elicitation phase) will evoke a much faster inflammatory response because abundant effector T cells may take information about the presence of the hapten onsite from not only LC or other dendritic cells but also from keratinocytes (non-professional antigen-presenting cells) [93,94,95]. The above-described mechanisms indicate where to look for possible molecular markers of ACD in future “omics” studies.

The word “contact” present in both terms—CA and ACD—implies that the skin is exposed to haptens via contact, i.e., from the outside. This is true in most cases, with one special exception being the systemic reactivation of ACD (SRACD) in which haptens enter the body not through the skin but via ingestion, inhalation, oral absorption, injection or implantation and are subsequently redistributed via circulation within the body, including the skin. Hematogenous ACD is a hybrid of “classical entrance” with SRACD. For details, see Table 4.

Throughout the medical literature, SRACD is commonly referred to as “systemic contact dermatitis”—a term that is easy to remember and pronounce but unfortunately misleading: The adjective “contact” implies that the triggering factor enters the skin from outside, while “systemic” implies a route from inside. Therefore, the term “systemic contact dermatitis” would literally mean “dermatitis caused by factors that enter the skin from inside yet at the same time from outside”, which is neither true in the induction phase (penetration from outside) nor in the elicitation phase (penetration from inside). The longer and more accurate term “systemic reactivation of allergic contact dermatitis” also stresses the fact that this type of reaction occurs in people previously sensitized via a typical route, i.e., skin contact. In the case of SRACD caused by systemic drugs, the synonym “symmetrical drug-related intertriginous and flexural exanthema” (SDRIFE) seems popular among authors [17,20,21]. Another somewhat historical yet still-used term is “baboon syndrome”, which is mainly synonymous with SRACD but for some authors is also synonymous with hematogenous ACD [96,97,98]. Systemic photoallergic dermatitis (see below) may also be viewed as a variant of SRACD.

People are constantly exposed to environmental chemicals, and chemical processes of hapten activation and the haptenization of autologous proteins into non-self-antigens are constantly ongoing in everyone’s bodies. Significant effort has been invested into the early identification of emerging haptens (e.g., new cosmetic ingredients, drugs and industrial chemicals) which have considerable potential to cause a CA and ACD [87,99,100]. However, the fact that only a minority of people will ultimately develop a CA to one or a few of the hundreds of haptens they are constantly exposed to makes it clear that the prerequisites for the development of a CA are individual predispositions in combination with other co-factors, e.g., inflammation on the site of hapten penetration [101,102,103]. The recognition of markers connected with the risk of developing a CA and specific molecular markers of ACD would benefit many people, either by facilitating prevention or improving the clinical diagnosis of ACD. Until the present, the few “omics” studies performed on ACD have focused on the haptenation of endogenous proteins, with an overall conclusion that different haptens target different proteins [104]. It seems that molecular markers suitable for the diagnosis of ACD have not yet been sought. Another promising avenue for future “omics” studies would be the study of whether the detection of haptenated proteins in the body might prove usable in clinical practice, e.g., as a laboratory method of detecting sensitized people or those at risk for sensitization.

### 4.7. Photoallergic Dermatitis

Photoallergic dermatitis is a variant of the above-discussed ACD, with the only difference being that the haptens initiating allergic reactions bind to endogenous proteins in a photochemical reaction. The external energy delivered by the photons is needed to activate photohaptens and enable them to bond covalently to endogenous proteins and change their spatial conformation from the tolerated “self” to the immunogenic “non-self” conformation. UVA is the typical energy carrier for most photohaptens, followed by UBV and visible light [105,106]. Photohaptens may enter the skin from outside, e.g., sunscreens and other cosmetic ingredients or topical drugs (photoallergic contact dermatitis) or from inside, e.g., systemic drugs and food additives or dietary supplements (systemic photoallergic dermatitis) [72,107]. Systemic photoallergic dermatitis may be viewed as a variant of SRACD as the photohaptens enter the skin from inside via circulation, with subsequent photohaptenization to form the actual antigens in the irradiated skin. At present, ketoprofen is by far the most frequent cause of photoallergic dermatitis in Europe, followed by other non-steroidal anti-inflammatory drugs and organic sunscreens [108]. A query of the literature returned no “omics” research on possible molecular markers of photoallergic dermatitis; however, due to shared pathomechanisms, such markers would likely be the same as in ACD.

### 4.8. Protein Contact Dermatitis (PCD)

PCD is an acquired inflammatory skin disease due to specific allergic reactions to foreign allergens—proteins with molecular weights exceeding 10,000 daltons which are usually of animal or plant origin [109]. An overview of confirmed causes of PCD is shown in Table 5. Due to their size, the allergens cannot penetrate healthy skin because they cannot cross an undamaged epidermis. Therefore, an inherent element of PCD is the disruption of the skin barrier manifesting as irregularities in the corneocytes—cells of the outermost layer of the epidermis, which contributes to its barrier function, as well as increased transepidermal water loss (TEWL) and decreased skin capacitance. After the resolution of visible dermatitis, the function of skin barrier remains impaired and recovers over several months, similar to AD [110]. Immune mechanisms overlapping type I and type IV allergic reactions have been postulated in PCD [109]. Proteins causing PCD are full antigens capable of inducing a type I reaction, which is in contrary to ACD in which haptens must bind to endogenous proteins to initiate an allergic reaction. On the other hand, the allergen-specific IgEs in PCD are bound as receptors on the surface of the LCs, whose participation resembles ACD, rather than a type I allergy. PCD is typically diagnosed as an occupational disease of farmers, veterinarians, butchers or food handlers—workers massively exposed to protein allergens in combination with wet work and repeated damages that disrupt the epidermal barrier and enable the penetration of large allergenic molecules into the skin [111]. The damage and inflammation that promotes PCD may also be due to preceding skin diseases, e.g., AD, ICD or ACD. A literature query returned no “omics” research that would single out candidate molecular markers of PCD.

### 4.9. Autoimmune Dermatitis

The concept of autoimmune dermatitis, initially referred to as “auto-sensitization” or “autoeczematization”, was first presented a century ago [112]. In the following decades, it remained a rather “niche” topic limited to clinical observations that a considerable group of children and adults with eczema react on patch or scratch tests with extracts from human dander, i.e., antigens from their own or donor keratinocytes, and develop skin reactions consistent with their disease (summarized in [113,114]). These early clinical observations were later supported by epidemiological data showing that patients with diagnosed AD are at a significantly higher risk of developing autoimmune diseases, with the highest odds ratio for alopecia areata (an 8–10-fold increased risk in AD patients), followed by vitiligo, celiac disease, inflammatory bowel disease, systemic lupus erythematosus, systemic sclerosis, thrombocytopenia and autoimmune thyroiditis [115,116]. Recently, AD has been announced somewhat emphatically as “a new autoimmune disease” [117]. Nevertheless, it is a matter of debate whether AD has always been an autoimmune disease without most clinicians and researchers realizing it or if perhaps new research methods have allowed a distinct disease within the heterogeneous group of patients thought to be ill with AD to be singled out. Most of the above-cited studies were carried out by researchers assuming that they were studying patients with AD. Taking into account the previously discussed vague diagnostic criteria and the heterogeneity of AD patients, it would be safer to say that in a subgroup of patients fulfilling the clinical criteria by Hanifin and Rajka, the disease seems to result from autoimmune reactions. The assumed mechanisms of the disease—an autoimmune reaction to the body’s own antigens but involving mechanisms typical of allergic reactions to exogenous antigens—makes autoimmune dermatitis a fascinating hybrid of allergy and autoimmunity. This speaks in favor of placing autoimmune dermatitis within the SoDE, yet as an entity separate from AD, regardless of the similarities in their clinical pictures. Two possibilities must be taken into account while dealing with this entity: autoimmune dermatitis may be secondary to preexisting AD and possibly other dermatitides, or it may emerge as a separate entity from the beginning.

The existence of autoimmune dermatitis as a separate disease seems to be supported by the recognized existence of a disease referred to as autoimmune progesterone dermatitis (AIPD). Compared with other dermatitides, AIPD becomes clinically distinct mainly through exacerbations synchronized with the menstrual cycle (catamenial pattern). The confirmed sensitizer here is progesterone—an endogenous steroid sex hormone whose molecular weight of 314 daltons places it among haptens. To date, more than a hundred cases diagnosed as AIPD have been reported throughout the medical literature; however, this number also includes cases of non-eczematous dermatoses, e.g., progesterone-triggered urticaria, which appears to comprise one-third of the reported cases [118]. The knowledge of AIPD pathomechanisms is limited. The combined evidence available thus far includes mentions of delayed-type reactions to progesterone on patch tests and intracutaneous tests (hinting at a type IV allergy) in patients with non-urticarial lesions, in addition to histopathologic picture featuring perivascular eosinophilic infiltrates with interface changes [119]. Another clue is a report of increased IFN-γ release by progesterone-exposed lymphocytes from a patient with erythema-annulare-centrifugum-like AIPD, hinting at a Th1-mediated response in the described case [120]. In future studies, true eczematous AIPD should be carefully selected from other progesterone-triggered skin diseases, e.g., autoimmune progesterone urticaria [121,122,123,124,125].

Autoimmune estrogen dermatitis (AIED) is a less well-known catamenial dermatitis with a handful of cases published so far; again, some of these studies seem affected by a too-liberal use of the term “dermatitis”. In a small group of patients with AIED, the formation of Langerhans cell nests in the epidermis and hair follicles, in addition to perivascular infiltrates of CD4+ and CD8+ lymphocytes in the dermis, was confirmed in skin lesions described as “prurigo, acneiform and annular erythema” but not in “urticaria-type AIED” [126] The characteristic feature of both AIPD and AEPD is the exacerbation of eczema in the luteal phase, especially on the days preceding menses, with remissions in the follicular phase [127]. A question remains as to whether similar mechanisms can be triggered in both women and men by other hormones but remain unrecognized because of a lack of periodicity in exposure. It seems that AIPD and AIED have not yet been the focus of “omics” studies.

### 4.10. Stasis Dermatitis

The main cause of stasis dermatitis is venous hypertension, which typically occurs in chronic venous insufficiency. Venous hypertension and the resulting blood congestion in the vessels (stasis) alone seem sufficient to cause stasis dermatitis. In a study of 10 patients with stasis dermatitis, all fully recuperated following a surgical intervention that resulted in the normalization of venous pressure [128]. Compression stockings were devised to prevent the ill effects of intravenous hypertension by applying counter-pressure from the outside. Among patients with stasis dermatitis who used compression stockings on an everyday basis, only 5% complained of frequent exacerbations of the disease compared with 64% of those who did not use the stockings [129].

Increased pressure in the veins and capillaries leads to an accumulation of leukocytes which attach to the endothelium (“leukocyte trapping”) and initiate processes leading to the apoptosis or necrosis of endothelial cells, fibroblasts, myocytes and parenchymal cells of the venous wall. The disruption in the vessels results in the extravasation of erythrocytes into surrounding tissues. Hemoglobin from these erythrocytes subsequently decomposes into hemosiderin, an insoluble complex of iron which induces an influx of macrophages attracted via hemoglobin scavenger receptor CD163. The accumulating macrophages secrete proteolytic enzymes and proinflammatory cytokines that induce skin damage which manifests as spongiosis in the epidermis, in addition to the fibrosis and neogenesis of capillary vessels in the dermis [130,131]. A macrophage mediator pivotal to stasis dermatitis is interleukin-31 (IL-31), which can directly evoke pruritus via the interleukin-31 receptor A (IL-31RA) present on peripheral nerves. The stimulation of IL-31RA on macrophages, basophils and keratinocytes may also augment ongoing inflammation [132]. Increased levels of IL-31 were also reported in AD; therefore, this marker seems to not be restricted to stasis dermatitis [132].

The distinctive molecular markers of stasis dermatitis should probably be sought among molecules related to venous hypertension and stasis rather than skin inflammation itself, though markers of macrophage activation and iron scavenging also seem promising with this regard. Compared to other dermatitides, stasis dermatitis seems rather easy to diagnose clinically thanks to the characteristic localization and coincidence of other signs of chronic venous insufficiency. On the other hand, chronic stasis dermatitis may be complicated by other SoDE diseases, most notably ACD [133].

### 4.11. Deficiency Dermatitis

The term “deficiency dermatitis” refers to a situation in which insufficient supplies of essential nutrients (vitamins, amino acids or microelements) lead to pathologic manifestations in the skin consistent with the clinical picture of eczema or dermatitis. Some cases may be due to a faulty processing of nutrients (e.g., genetic defects) rather than an insufficient supply. Clinical cases of deficiency dermatitis may pose significant challenges to doctors and researchers because patients may suffer from combined nutrient deficiencies, or deficiencies may aggravate the course of preexisting diseases from the SoDE.

Zinc deficiency dermatitis is arguably the most studied form of deficiency dermatitis. Zinc is a microelement substantial for the differentiation of keratinocytes, as well as for anti-inflammatory and wound healing processes. Its deficiency leads to ATP-mediated inflammation and impairs the maturation and functioning of T, B and NK lymphocytes [134]. Zinc deficiency may develop in the course of acrodermatitis enteropathica, which is an autosomal recessive genetic disease. An acquired zinc deficiency may be due to an insufficient supply, malabsorption or excessive loss of zinc, e.g., in disorders of the gastrointestinal or urinary tract, or increased zinc requirement (e.g., pregnancy or breastfeeding) [135]. A full-blown zinc deficiency manifests clinically with the classical triad of dermatitis, alopecia and diarrhea. In cases of isolated zinc deficiency, zinc supplementation brings about a rapid recovery [136]. An acquired zinc deficiency may also contribute to other nutritional deficiencies, e.g., niacin (vitamin B3) and biotin (vitamin B7) deficiencies.

The role of vitamin D deficiency in initiating and aggravating the course of inflammatory dermatoses is well documented; the same is true for the beneficial effects of its supplementation [137,138,139]. However, low vitamin D levels could hardly be viewed as a nutritional deficiency because “vitamin D” is in fact a group of nuclear hormones with pleiotropic actions, including immunomodulation [140,141,142]. Under favorable circumstances, all active forms of vitamin D are synthesized entirely in the body; therefore, low levels of vitamin D should be viewed as a hormonal disorder rather than nutrient deficiency [143]. Hints as to other deficiencies that might induce or aggravate dermatitis seem mainly based on accidental clinical observations, uncontrolled studies or animal experiments. Nevertheless, they certainly deserve further research, including “omics” studies into the molecular markers of predisposition and active disease. The present practice of measuring nutrient levels in blood may prove insufficient due to a gap between population norms and the actual individual demand, as well as between blood levels and tissue bioavailability.

## 5. Discussion

As summarized above, AD has been the almost exclusive topic of “omics” research to date, with other diseases from the SoDE hardly ever studied. Most studies in this field compare purported molecular markers in AD patients versus healthy controls but not with other diseases from the spectrum. In this regard, it seems quite symptomatic that a patent was recently submitted for a transcriptomic method of diagnosing AD based on a research study that did not take into account any of the common dermatitides which would require a differential diagnosis against AD in everyday clinical routine [58]. Therefore, it cannot be excluded that a range of other inflammatory dermatoses would be classified by the submitted test as “atopic dermatitis” without actually being AD. With the current state of knowledge, it is difficult to predict which molecular markers would prove specific to particular diseases. It seems probable that many markers of inflammation, skin damage and repair singled out thus far in AD research are actually expressed throughout the entire SoDE. The very rare “omics” studies comparing AD with other dermatoses (e.g., the last three entries in Table 3) have shown some differences; unfortunately, none of the compared diseases were from the SoDE. A study of sweat glucose and GLUT2 expression that compared results from AD patients with a heterogeneous group of six subjects, each with another skin disease, did not reveal any marker that would single out AD [144]. Therefore, it is difficult to assess how many markers identified so far in the “omics” studies in AD patients are indeed specific to AD and how many are present in other diseases from the SoDE or beyond the spectrum.

## 6. Conclusions

Diseases from the SoDE share many clinical features and they may coexist, making an accurate diagnosis very challenging for clinicians. Some diagnostic criteria currently used seem vague, overlapping, ill-defined or misleading. In order to fill in the gap, large-scale “omics” research of all diseases from the SoDE is urgently needed to identify possible molecular markers that might support everyday clinical diagnoses. The author’s prediction is that within a decade, results of “omics” research can indeed change clinical practice by redrawing the boundaries between known dermatitides and probably also by singling out new nosologic entities with unique molecular markers. The provision for this success is that the research efforts will expand and cover all diseases from the spectrum.

## 7. Future Directions

In order to enable the widespread use of molecular markers in routine diagnostics, non-invasive sampling techniques should be favored, e.g., skin swabs or tape-stripping, saliva, sweat, feces or urine samples and possibly blood samples. Skin biopsy might be difficult to carry out routinely, especially in small children. Should future research prove that skin biopsies are unavoidable for an accurate diagnosis, molecular markers should be preferred that are present in both lesional and non-lesional skin. Such markers would probably be less susceptible to bias via the choice of a “wrong” biopsy site; moreover, a skin biopsy that could be performed at any site would arguably be more acceptable for patients who could opt for a biopsy site easy to tend and conceal. Molecular markers that are detectable regardless of the current disease activity—from complete remission to erythroderma—would be more convenient for the diagnosis, whereas markers with expression proportional to disease activity would be valuable in the monitoring of disease progression and the effectiveness of treatment.

The desirable response to everyday challenges of differential diagnosis would be a systematic comparative study of all diseases from the SoDE and other inflammatory dermatoses. While undertaking the postulated research, researchers should be fully aware of the fact that the present clinical diagnostic criteria of the diseases from the SoDE are vague and overlapping; therefore, study populations selected on this basis must be considered heterogeneous a priori and not representative of any specific disease. A favorable approach to this problem might be an inclusive rather than exclusive recruitment of eczema patients and the “reverse-engineering” of obtained results, i.e., the division of the study population into the most homogenous clusters according to molecular markers and a subsequent search within the clusters for distinctive clinical features including symptoms, severity, natural course, response to treatment modalities, etc. The proposed approach is similar to the studies discussed above [47,48,49], with the substantial difference of including patients from the entire SoDE along with healthy controls and preferably also patients with other skin diseases. Possible gender, age and racial/ethnic differences should also be taken into account while redefining the boundaries between the entities within the spectrum [39,40,41]. Future researchers should also bear in mind that diseases from the SoDE may be mimicked by genodermatoses, other genetic syndromes with skin involvement, paraneoplastic syndromes and adverse drug reactions—a diverse group of diseases, some of which may share clinical appearances, mechanisms and molecular markers with diseases from the SoDE [145,146,147,148,149]. 

Applying “omics” methodology to the entire SoDE might result in a rearrangement of the diseases into more coherent entities diagnosed reliably via validated molecular and clinical criteria. The ambitious goal of collecting “omics” results sufficient to elaborate better diagnostic criteria and systematize the entire SoDE in its diversity would require a harmonized collaboration of dermatology and allergy societies, departments and research centers in many countries and continents, expanding the concept and scope of already existing registries devoted to AD [150,151]. Such action would ultimately enable a more accurate diagnosis, resulting in more effective disease management and better quality of life for patients.

## Figures and Tables

**Table 1 ijms-24-10468-t001:** A proposed systematics of skin diseases from the spectrum of dermatitis and eczema (SoDE).

Etiopathology Classes	Examples of Clinical Diagnoses ^1^
No known/detectable triggers	Intrinsic atopic dermatitis (endogenous eczema)
Triggered by exogenous factors without the involvement of specific hypersensitivity	Irritant contact dermatitis Winter dermatitis (hand, foot, and generalized) Microtrauma dermatitis Friction dermatitis Phototoxic dermatitis Radiodermatitis (radiation dermatitis) Seborrheic dermatitis
Triggered by exogenous factors with the involvement of specific immunological hypersensitivity	Extrinsic atopic dermatitis (exogenous eczema) Allergic contact dermatitis SDRIFE/SRACD ^2^ Photoallergic dermatitis Protein contact dermatitis
Autoimmune reactions	Autoimmune dermatitis, including: Autoimmune progesterone dermatitis Autoimmune estrogen dermatitis
Homeostatic imbalance	Stasis dermatitis Deficiency dermatitis

^1^ Some clinical diagnoses may overlap, while others may be divided between classes because of their heterogeneity. ^2^ SDRIFE (symmetric drug-related intertriginous and flexural exanthema) and SRACD (systemic reactivation of allergic contact dermatitis) are both variants of ACD in which haptens enter the body through a non-cutaneous route (ingestion, inhalation, or injection) and are further distributed via blood circulation into the skin where they ultimately cause an allergic reaction [17,20,21].

**Table 2 ijms-24-10468-t002:** Denominations of diseases from the spectrum of dermatitis and eczema (SoDE) that essentially are not diagnoses, though their use may sometimes prove practical in clinical practice.

Customary Term	Reason for Use
Airborne dermatitis	Specific route of exposure.
Asteatotic eczema (eczema craquelé; asteatotic dermatitis)	Characteristic clinical picture: skin fissures occurring in irregular, polygonal or curvilinear patterns, referred to as “crazy paving”.
Connubial dermatitis (consort dermatitis; dermatitis by proxy)	Specific route of exposure (allergic symptoms in a seemingly non-exposed person due to transmission of haptens from an exposed person in close surroundings, typically a member of the same household).
Dyshidrotic eczema (dyshidrotic dermatitis, pompholyx)	Characteristic clinical picture (vesicles or bullae) and localization (palms and soles); persistence and recurrence.
Foot dermatitis (foot eczema)	Characteristic localization, similarity of clinical picture regardless of etiopathogenesis, frequent polyetiology, negative impact on daily routines and work and resistance to treatment.
Hand eczema (hand dermatitis)	Characteristic localization, similarity of clinical picture regardless of etiopathogenesis, frequent polyetiology, negative impact on daily routines and work and resistance to treatment.
Nummular eczema (nummular dermatitis)	Characteristic clinical picture, persistence and recurrence.
Occupational dermatitis (occupational eczema)	Specific exposure, legal status and economic impact.

**Table 3 ijms-24-10468-t003:** A summary of “omics” studies of AD in humans from the last 5 years—an update to the overview found in Afghani et al. (2022).

Study Description	Selected Results of Possible use in Differential Diagnosis	Comment	Ref.
Transcriptomic analysis of blood and skin samples	The upregulation of 69 genes and the downregulation of 35 genes in the blood samples from AD patients when compared with controls. The strongest levels of mRNA upregulation in the blood were observed for CCL23 (MIP-3), indoleamine-pyrrole-2,3-dioxygenase (IDO)-1, IL-5 receptor subunit IL5RA, IL-33 receptor IL1RL1 (ST2), histamine receptor 4 (HRH4) and CCR3. The upregulation of 2070 genes and the downregulation of 2591 genes in the lesional skin from AD patients when compared with controls. The upregulation of 635 genes and the downregulation of 646 genes in the non-lesional skin from AD patients when compared with controls.	North American children (28 AD; 18 HC) Not compared with other diseases from the SoDE	[51]
Transcriptomic and proteomic analysis of skin and blood samples	Out of 354 proteins from the inflammation, CVD II, CVD III and neuroimmunology panels, 161 were differentially expressed in lesional AD skin and 69 in nonlesional AD skin when compared with skin samples from controls. The expression of these markers in AD skin was much higher than their expression in blood.	North American adults (20 AD; 28 HC) Not compared with other diseases from SoDE	[52]
Proteomic analysis of skin tape strips	Out of 371 proteins from the inflammation panel, 45 were significantly higher in strip samples from AD patients when compared with samples from controls. The collective protein levels were correlated with total and food-specific IgE levels, the number of skin lesions and transepidermal water loss (TEWL).	North American children (21 AD with food allergy, 19 AD without food allergy and 22 HC) Not compared with other diseases from SoDE	[53]
Proteomic analysis of suction blister fluid	The upregulation of IL-13 in TH2 and NKT cells, the upregulation of IL-22 in TH22 and NKT cells, the upregulation of IL26 in NKT cells, the upregulation of CLEC7A, amphiregulin/AREG and EREG in dendritic cells and the upregulation of CCL13 in macrophages from the suction blister fluid cells of AD patients when compared with controls.	Caucasian adults (13 AD; 10 HC) Not compared with other diseases from SoDE	[54]
Multi-omics analysis of feces	The expression of EARS2 (gene encoding a mitochondrial glutamyl-tRNA synthetase) in fecal samples indicated AD patients with a 100% selection probability.	Asian children (38 AD; 46 HC) Not compared with other diseases from SoDE	[55]
Transcriptomic analysis of skin samples	Th1- and Th22-related mRNA expression was significantly higher in AD skin lesions than in skin samples from controls. Th2- and Th17-related mRNA and IL-17 signature expression levels were significantly higher in skin samples from AD lesions than in non-lesional AD skin samples and skin samples from controls.	African American adults (6 lesional AD, 6 non-lesional AD and 6 HC) Not compared with other diseases from SoDE	[56]
Proteomic analysis of skin swabs	Swabs from non-lesional skin revealed significantly lower levels of 18 proteins, serpin B6, carboxypeptidase M, 20S core proteasome and its adaptors (PSMA2, PSMA5, PSMB2, PSMB4 and KCTD5), tissue alpha-L-fucosidase (FUCA1), zinc-alpha-2-glycoprotein (AZGP), N-acetylgalactosamine-6-sulfatase (GALNS), 7-dehydrocholesterol reductase (DHCR7), ester hydrolase (C11orf54), phospholipase B (PLB1; PLBD2), phospholipase D3 (PLD3), phospholipid-binding annexin A7 (ANXA7), acid phosphatase (ACPP) and nucleophosmin (NPM1), and an increased expression only of the protein TDRD15 in AD patients when compared with healthy controls.	Caucasian adults (8 AD; 8 HC) Not compared with other diseases from SoDE	[57]
Transcriptomic analysis of skin surface lipids	Lower expressions of mRNAs related to keratinization (LCE, PSORS1C2, IVL and KRT17), triglyceride synthesis and storage (PLIN2, DGAT2 and CIDEA), wax synthesis (FAR2), ceramide synthesis (GBA2, SMPD3 and SPTLC3), antimicrobial peptides (DEFB1) and intercellular adhesion (CDSN) in facial sebum specimens from AD patients when compared with healthy controls. Higher expression of mRNA coding a Th2-cytokine CCL17 in facial sebum specimens from AD patients when compared with healthy controls. KRT17 and CCL17 expression levels in facial sebum specimens from AD patients significantly correlated with the severity of the disease (EASI score).	Asian children (16 AD; 23 HC) Not compared with other diseases from SoDE	[58]
Proteomic analysis of serum samples confirmed via ELISA and immunohistochemistry	Significantly elevated expression of serum Cofilin-1 in AD patients when compared with healthy controls.	Asian children and adults (45 AD; 45 HC) Not compared with other diseases from SoDE	[59]
Proteomic and transcriptomic analyses of saliva samples	The ratio of Th1 to Th2 cytokines (IL-8/IL-6) was significantly higher in saliva samples from infants with AD when compared with healthy controls. Saliva levels of microRNA miR-375-3p were significantly lower and levels of miR-21-5p were significantly higher among infants with AD when compared with healthy controls. The alpha diversity of bacterial RNA expression (Simpson index) in saliva was significantly higher among infants with AD when compared with healthy controls. Proteobacteria RNA levels were significantly higher in saliva from infants with AD when compared with healthy controls.	North American infants (37 AD; 92 HC) Not compared with other diseases from SoDE	[60]
Proteomic analysis of skin samples	The upregulation of the proteins S100A7, KRT16, S100A9, S100A8 and SPRR2D and the downregulation of F2, STEAP4, HP, EPHB3 and HRAS were observed in keratinocytes from AD non-lesional skin samples when compared with samples from healthy controls. The upregulation of the proteins KRT16, S100A7, S100A9, SERPINB4 and S100A8 and the downregulation of H2BU1, KRT77, CA13, FHL1 and LTF in keratinocytes from AD lesional skin samples when compared with samples from healthy controls.	Caucasian adults (10 AD; 10 HC) Not compared with other diseases from SoDE	[61]
Metabolomic analysis of skin and blood samples	Nineteen metabolites and their ratios differed significantly between AD lesional and psoriatic lesional skin. Only one metabolite (acetylcarnitine) was significantly more abundant in non-lesional skin from psoriasis patients when compared with non-lesional skin from AD patients. Five metabolites (citrulline, glutamate, proline, carnitine and octadecenoylcarnitine) were significantly higher in serum samples from psoriasis patients when compared with AD patients.	Caucasian adults (skin biopsies: 15 AD, 20 PS, 17 HC. Serum: 25 AD, 55 PS and 63 HC) Comparisons AD vs. psoriasis (not a SoDE disease) Healthy controls mentioned, but no results shown	[62]
Transcriptomic study of skin biopsies	Out of 15,719 differentially expressed genes (gene up- or downregulated), 1838 were exclusively found in AD, 2879 in psoriasis and 6275 in prurigo nodularis. Prurigo nodularis and psoriasis shared 3775 differentially expressed genes, prurigo nodularis and AD shared 1551 differentially expressed genes, psoriasis and AD shared 1565 differentially expressed genes and all three dermatoses shared 1082 differentially expressed genes. Prurigo nodularis and psoriasis shared the most Th17/22-related differentially expressed genes, including CXCL1, DEFB4A, LCN2, PI3, IL8, S100A7/8/9/12 and SERPINB1/4. Prurigo nodularis and AD shared CXCL1/2, PI3, S100A7/8/9 and SERPINB1/4.	North American adults (100 AD, 100 PS and 100 PN) AD compared with skin diseases not belonging to SoDE	[63]
Transcriptomic and epigenomic study of skin biopsies	Upregulated gene expression in the skin samples for NF-κB1, NF-κB2, Protein kinase C-binding protein NELL2, artemin, nerve growth factor (NGF) and GDNF family receptor alpha-1 (GFRα1) but not IL-4 in AD patients when compared with healthy controls. No markers specific for particular diseases; qualitative rather than quantitative differences between the diseases.	Caucasian adults (45 CNPG, 40 BRP, 40 AD and 45 HC) AD compared with skin diseases not belonging to SoDE Possible bias: CNPG may be related to AD [64]	[65]

AD—atopic dermatitis; BRP; brachioradial pruritus; CNPG; chronic nodular prurigo; HC—healthy controls; PN—prurigo nodularis; PS—psoriasis; SoDE—spectrum of dermatitis and eczema.

**Table 4 ijms-24-10468-t004:** Clinical forms of allergic contact dermatitis (ACD).

Clinical Form	Route of the Hapten Exposure
ACD limited to the site of contact	Direct skin contact with the sensitizing hapten.
ACD with local spreading	After entering into the skin, the hapten disperses from the primary focus via lymph vessels, resulting in the emergence of secondary foci (satellites) in close vicinity.
Hematogenous ACD	After passing through the skin in the exposed site (primary patch), the hapten enters systemic circulation via lymph vessels or veins and is redistributed to the skin (and other organs) via arteries. A primary patch can be found with subsequent secondary eczematous foci dispersed over the body, with possible predominance in flexures and skin folds. Involved areas may also be limited to previous localizations of ACD. As the route includes the portal vein and the liver, actual haptens may be metabolites of the substance entering the body; in such cases the primary patch may be absent.
Systemic reactivation of ACD (SRACD)	The (pro)hapten enters systemic circulation directly (via inhalation, oral absorption, injection or implant) or indirectly via the gut and liver passage and is further redistributed to the skin and other organs via the arteries. Actual haptens may be metabolites of the substances entering the body. The eczematous foci may be dispersed over entire skin areas, show a predominance in flexures and skin folds or be limited to previous locations of ACD. The primary patch is not present in SRACD.

**Table 5 ijms-24-10468-t005:** Causes of protein contact dermatitis (PCD).

Protein Group	Examples (Documented Cases)
Plant proteins, food	Flour and dough (corn, wheat, rye, barley and buckwheat) Fruit (apple, banana, fig, kiwi, lemon, lime, orange, peach, pear and pineapple) Herbs (chamomile, chicory, coriander and hops) Nuts (almond, hazelnut, peanut and pecan) Spices (allspice, caraway, cinnamon, clove, curry, ginger, paprika, parsnip and parsley) Vegetables (asparagus, aubergine, beans, broccoli, cabbage, carrot, cauliflower, celery, coriander, cucumber, endive, garlic, horseradish, komatsuna, leek, lettuce, maize, melon, onion, onion weed, paprika, potato and tomato, watercress)
Plant proteins, non-food	Corn starch Flowers and ornamental plants (tulip, lilac, lily, ficus, chrysanthemum and spathe) Latex Plant parts, nonedible (cucumber leaf, castor bean, grain dust, hay and straw) Wood (sapele and western red cedar)
Animal proteins, food	Meat (beef, chicken, horse, lamb, mutton and pork) Edible offal (blood, heart, intestines, liver, mesenteric fat, stomach, suet, sweetbreads and tallow) Milk and dairy products (cheese and egg) Fish (butterfish, cod, eel, hake, herring, salmon, tilapia, tuna fish and whiting) Shellfish (cuttlefish, lobster, crab, scallop, shrimp and squid)
Animal proteins, non-food	Animal dander and hair (cow, deer, giraffe, horse and pig) Animal excreta and bodily fluids (urine, semen and amniotic fluid) Arthropods (house dust mites and Tetranychus urticae) Insects and larvae (Chironomus thummi thummi midge larvae, cockroaches, crickets, Galleria mellonella larvae, processionary caterpillars, maggots and Zophobas morio superworms) Parasites (Anisakis)
Fungal proteins, food	Mushrooms (shiitake) Molds
Fungal proteins, non-food	Molds
Protein hydrolysates	Hydrolyzed wheat protein
Bioengineered proteins	Enzymes (alpha-amylase, lactase, peroxidase and purple acid phosphatase)

## Data Availability

Data sharing not applicable.

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
