# Peer review of "Diseases from the Spectrum of Dermatitis and Eczema: Can “Omics” Sciences Help with Better Systematics and More Accurate Differential Diagnosis?"

_ijms, 2023, doi:10.3390/ijms241310468_

Round 1
Reviewer 1 Report
Spiewak wrote an impressive review work on unmet needs and definition of eczema and dermatitis, highlighting the inflation of work about atopic dermatitis against other forms of eczema. The author propose omics approach to solve the flaws of modern research on this field. The paper could be approved in is present form
Author Response
Thank you very much indeed for your very favorable assessment.
Reviewer 2 Report
Reviewer comments and suggestions
The authors in this study presented this review as a systematics of diseases that are spectrum of dermatitis and eczema, based on the origin of causative factors and pathomechanisms involved. Problems in differentiating between these diseases are deliberated along with considerations, to what extent the advances in omics sciences might help to overcome them. The authors suggested that with the help of omics study, more than 90% of published papers were devoted to atopic dermatitis with a striking underrepresentation of other diseases from the spectrum of dermatitis and eczema that collectively exceed by far the rates of atopic dermatitis. A clinics-oriented approach is proposed for future omics studies in the field of dermatitis and eczema
Overall, the manuscript was well written. However, a few concerns/comments needed to be explained/modified.
- I could not find reference 1, please add it.
- Line 60-64 The lines needed appropriate reference
- Line 88-89 Which study the authors are discussing, please mention it
- Line 110-114 Did the authors use some valid studies for proposing these criteria, please mention as well
- Line 154-159 it would be nice if the authors were clear about the method and analysis of this study
- Tables 4 and 5 The tables did not look professional, please modify
- It would be nice if they can draw some figures regarding these interactions or associations for better understanding
- All reference formats should be modified based on MDPI guidelines
Author Response
I sincerely thank the Reviewer for the review and thoughtful comments.
Below are the specific responses:
- I could not find reference 1, please add it.
The reference #1 was in a motto that was deleted by the editorial staff according to the publisher policy. The references have been reordered to correspond with the present state.
- Line 60-64 The lines needed appropriate reference
Re: Respective references were added.
- Line 88-89 Which study the authors are discussing, please mention it
Re: Most, if not all, clinical studies in the field are flawed by imprecise clinical criteria used at the recruitment of study participants. It seems that listing virtually all "omics" human studies with inclusion criteria based on Hanifin and Rajka criteria or their modifications would not add much merit while greatly expanding the reference list. Instead, the reader is directed to examples discussed later on in the paper.
- Line 110-114 Did the authors use some valid studies for proposing these criteria, please mention as well
Re: The proposed classification is an attempt at ordering known diseases from the SoDE according to their presumed etiopathology. Admittedly, the division is as arbitrary as ordering the diseases by involved regions or alphabetically - none of such orders seems possible to validate, as these features are not intended as diagnostic criteria. The introduction to the proposed systematics has been rewritten to clarify this.
- Line 154-159 it would be nice if the authors were clear about the method and analysis of this study
Re: This part was intended to indicate that the following discussion will not cover all aspects of AD but will focus on topics that seem relevant to the paper, and to indicate some examples of recent articles on AD as a further reading. In order to make it clear that there are many more relevant papers in the field, a statement "to name just a few recent examples" has been added.
- Tables 4 and 5 The tables did not look professional, please modify
Re: The tables were made using the template provided by the publisher. I tried and made the best possible within the limits of the template.
- It would be nice if they can draw some figures regarding these interactions or associations for better understanding
Re: Following your suggestion, a graphical abstract has been added to visually demonstrate the main idea of the paper.
- All reference formats should be modified based on MDPI guidelines
Re: The references have been re-checked and seem to comply with the journal guidelines.
Thank you very much indeed for your time and input invested into the paper.
Radoslaw Spiewak
Reviewer 3 Report
Dear author,
I have studied with great interest the manuscript ‘’ Diseases from the spectrum of dermatitis and eczema: Can omics sciences help with better systematics and more accurate differential diagnosis?’’
The manuscript is adequate, dealing with a novel and interesting topic on dermatitis spectrum diseases. The text is clear, well organized, and interesting to the reader. The references are appropriate and updated. The tables correspond to the description in the text, are well designed and reflect important information.
I have some comments:
-Abbreviations should be described the first time they appear in the text. For example, atopic dermatitis (AD) should be described on line 28 of the manuscript and from that line onwards the abbreviation should be retained whenever AD is referred to (this can be changed in the headings and retain Atopic Dermatitis). Please apply to all abbreviations.
- As the title of your article is a question, I would recommend answering this question in the conclusion section.
-The idea of complex differential diagnosis is very interesting, perhaps if the dermatological associations of the different countries would collaborate with the hospitals and make common databases where all the information of patients with dermatitis appears, it could help the knowledge of this disease and facilitate phenotypic classification studies (Example, PMID: 36935039, PMID: 37172898).
-On the other hand, it is not clear to me what type of publication is this study, it could be a narrative review or a systematic review. However, in order to consider it a systematic review it would be necessary to include what criteria were used to select the articles included. I would also be missing a flowchart of the literature search and all requirements of PRISMA checklist for systemic reviews.
Author Response
I sincerely thank the Reviewer for the favorable review and comments. Below are the responses to the particular points:
-Abbreviations should be described the first time they appear in the text. For example, atopic dermatitis (AD) should be described on line 28 of the manuscript and from that line onwards the abbreviation should be retained whenever AD is referred to (this can be changed in the headings and retain Atopic Dermatitis). Please apply to all abbreviations.
Re: The abbreviations have been ordered in line with the Reviewer's suggestion. Full terms were preserved in the abstract and tables (which are kind of separate entities and should be self-explanatory), titles of sections, as well exact quotes (to be true to the original) and while discussing terminology (to ensure easier understanding). All remaining terms were replaced with abbreviations.
- As the title of your article is a question, I would recommend answering this question in the conclusion section.
Re: Following the Reviewer's suggestion, I tried to answer and predict the future in the last sentence of conclusions.
-The idea of complex differential diagnosis is very interesting, perhaps if the dermatological associations of the different countries would collaborate with the hospitals and make common databases where all the information of patients with dermatitis appears, it could help the knowledge of this disease and facilitate phenotypic classification studies (Example, PMID: 36935039, PMID: 37172898).
Re: Thank you very much for this valuable idea that has been incorporated into the last paragraph of the paper.
-On the other hand, it is not clear to me what type of publication is this study, it could be a narrative review or a systematic review. However, in order to consider it a systematic review it would be necessary to include what criteria were used to select the articles included. I would also be missing a flowchart of the literature search and all requirements of PRISMA checklist for systemic reviews.
Re: The paper is a narrative review which is now clearly stated in the abstract.
I thank you very much for your time and valuable suggestions.